# Porcine Model of the Growing Spinal Cord—Changes in Diffusion Tensor Imaging Parameters

**DOI:** 10.3390/ani13040565

**Published:** 2023-02-06

**Authors:** Karolina Barbara Owsińska-Schmidt, Paulina Drobot, Anna Zimny, Marcin Adam Wrzosek

**Affiliations:** 1Department of Internal Medicine and Clinic for Horses, Dogs and Cats, The Faculty of Veterinary Medicine, Wrocław University of Environmental and Life Sciences, 50-366 Wrocław, Poland,; 2Department of General Radiology, Interventional Radiology, and Neuroradiology, Wrocław Medical University, 50-556 Wrocław, Poland

**Keywords:** diffusion tensor imaging, magnetic resonance imaging, spinal cord, animal models, porcine, growth process

## Abstract

**Simple Summary:**

Spinal cord injuries are a great concern in veterinary and human medicine. The main problem in this field is the difficulty of evaluating the degree of damage objectively using standard structural imaging methods. In our work, we used an advanced imaging technique that has promising applications for the objective assessment of the microstructure of the spinal cord. We decided to apply this method to healthy pigs as a model organism due to their rapid weight gain and anatomical and physiological similarity to humans. We obtained results that could be useful in determining reference values for the undamaged spinal cords of animals and growing humans. The obtained values related to porcine growth will allow us to achieve a model of the growing spinal cord that can be used in both human and veterinary medicine for the objective assessment of the microstructure of the spinal cord.

**Abstract:**

Diffusion tensor imaging (DTI) is an advanced magnetic resonance imaging (MRI) technique that has promising applications for the objective assessment of the microstructure of the spinal cord. This study aimed to verify the parameters obtained using DTI change during the growth process. We also wanted to identify if the DTI values change on the course of the spinal cord. The model organism was a healthy growing porcine spinal cord (19 pigs, Polish White, weight 24–120 kg, mean 48 kg, median 48 kg, age 2.5–11 months, mean 5 months, median 5.5 months). DTI parameters were measured in three weight groups: up to 29 kg (five pigs), 30–59 kg (six pigs), and from 60 kg up (eight pigs). DTI was performed with a 1.5 Tesla magnetic resonance scanner (Philips, Ingenia). Image post-processing was done using the Fiber Track package (Philips Ingenia workstation) by manually drawing the regions of interest (nine ROIs). The measurements were recorded for three sections: the cervical, thoracolumbar and lumbar segments of the spinal cord at the C4/C5, Th13/L1, and L4/L5 vertebrae levels. In each case, one segment was measured cranially and one caudally from the above-mentioned places. The values of fractional anisotropy (FA) and apparent diffusion coefficient (ADC) were obtained for each ROIs and compared. It is shown that there is a correlation between age, weight gain, and change in FA and ADC parameters. Moreover, it is noted that, with increasing weight and age, the FA parameter increases and ADC decreases, whereas the FA and ADC measurement values did not significantly change between the three sections of the spinal cord. These findings could be useful in determining the reference values for the undamaged spinal cords of animals and growing humans.

## 1. Introduction

Spinal cord injuries (SCIs) are of great concern in human and veterinary medicine. The main problem in this field is the difficulty of objectively evaluating the degree of SCI using standard structural magnetic resonance images (MRI) [1]. Assessing the prognosis for recovery or the degree of spinal cord regeneration after an applied treatment is still a clinical challenge in neurology and radiology [2,3,4]. Research shows that these two aspects are extremely important for patients with SCI [5], as initial values of objective assessment such as DTI for the status of the healthy spinal cord before the injury are lacking.

Diffusion tensor imaging (DTI) is an advanced MRI technique that has promising applications for evaluating microstructure and, indirectly, the function of the spinal cord [6,7,8,9,10,11]. This method is predicted to be especially useful in patients with SCIs. It has the potential to allow for a more accurate classification of damage, prognosis, and assessment of recovery and treatment than conventional techniques (standard MRI) in both adult and pediatric patients [1,12,13,14,15,16,17,18].

DTI is based on the in vivo measurement of water diffusion in tissue. It is founded on the fact that diffusion is not chaotic, but limited and directional. This directional dependency is known as anisotropy. It is measured in different directions, and the mean determines the diffusion tensor—a mathematical object that fully describes the dependence of diffusion on its orientation. When a diffusion tensor is known, it is possible to quantify anisotropy. The comparison of numerical values, obtained as a result of imaging by this method, alongside others (e.g., fractional anisotropy (FA) and apparent diffusion coefficient (ADC)), allows for an assessment of the spinal cord’s microstructure [6,16,19,20].

Large animal models of spinal cord DTI parameters are very limited. Most often, a rat model of an intact or damaged spinal cord is reported [8,12,15,21,22]. Some studies describe dogs affected by intervertebral disc herniation (IVDH) as a large animal model of SCI (Dachshunds and mixed-breed dogs are the most common), while others use healthy Beagle dogs as exemplar undamaged spinal cord models [23,24]. Another report applies DTI to normal-appearing spinal cords of 13 dogs of different breeds in two locations (cervical and thoracolumbar) [25].

We decided to use pigs without SCI as a model organism due to their rapid weight gain and anatomical and physiological similarity to humans. In our work, we included in the analysis both the weight and age of the examined pigs to better illustrate the rapid growth time of the selected animal model. These features enable quick and reliable results with different sizes during the growth of the animal’s spinal cord, forming an excellent model for translational SCI research in the fields of veterinary and human medicine. Porcine models offer an alternative not only because of their anatomical and physiological similarities to humans, but also the availability of genomic, transcriptomic, and progressive proteomic tools for the analysis of porcine species [26]. Moreover, the size of this animal model allows the use of the same imaging tools, scanners, software, and interpretation used in everyday clinical and advanced imaging sets. Despite these advantages, there are no reports on the use of these animals as a model for undamaged, growing spinal cords, which proves the novelty of our research.

Many reports describe DTI values in healthy people (children and adults) with an undamaged spinal cord, evaluating differences in age and cord region. However, the obtained results are still not clear and well-described as compared to the DTI parameters of the brain [27,28,29,30,31].

The goal of this study was to assay the change in individual parameters obtained in the growth process (dependence on age and body weight) and identify the eventual statistically significant differences in these values between individual sections of the spinal cord.

The purpose of this study was to emphasize the benefits of creating and optimizing animal translational models. The results are thought to contribute to the determination of DTI reference values for pigs with intact spinal cords and could be used for comparison with the ones obtained from porcine models of SCI in further studies. The study verifies the pig as a suitable model for translational medicine in this field. DTI is a very promising method; however, it is quite a new technique for clinical use. Obtaining objective reference values is necessary for further clinical trials in order to have a reference point on the values obtained from clinical patients with SCI.

## 2. Material and Methods

### 2.1. Animal Model

Piglets of the same breed (Polish White) with similar body weight, i.e., approx. 15–20 kg, were used in the study. The animals were observed for about 4 months, and in one case for 9.5 months. The exact date of birth of the animals was not known, because they came from a breeding farm. Therefore, the age was estimated based on the knowledge of the typical weaning age of animals of this breed and the time of their observation in the study. Local Ethics Committee approval for the research was obtained (87/2017). To guarantee the protection and welfare of the animals participating in the study, the 3Rs principle (replacement, reduction, and refinement) was applied [32,33].

Each animal had at least two weeks to calmly acclimatize before testing (the so-called “handling” procedure). During this time, daily observation of the animals for clinical status was carried out. Animals were accustomed to contact with employees to limit stressors during the research. No clinical signs of disease were observed in any animal during the acclimatization period.

All animals were kept, according to the defined earlier environment, in 3 m × 1.70 m boxes with high-sawdust bedding to ensure appropriate conditions for development and growth without limiting the motor space of each animal while ensuring optimal conditions for socialization between animals throughout the observation period (Figure 1).

The main advantage of using the DTI method is the in vivo assessment of spinal cord microstructure. Therefore, it is necessary to study animal models, and there is no possibility of replacing them with in vitro tests on tissue or cell lines. Since DTI is a non-invasive procedure, values were taken from the MRI control tests performed in the above-mentioned research project. The number of animals qualified for the study was estimated to ensure the reliability and validity of the obtained outcomes. The study was conducted in 2017–2020, and some results were analyzed retrospectively.

A total of 19 healthy pigs, aged 2.5–11 months, qualified for the research. DTI parameters were measured in three weight groups: up to 29 kg (*n* = 5), 30–59 kg (*n* = 6), and from 60 kg up (*n* = 8).

### 2.2. Anesthesia

The examinations were performed under general anesthesia. In each case, the clinical condition of the animal was assessed before the procedure. Intramuscular premedication with 20 µg/kg medetomidine (Cepetor^®^, CP-Pharma, Burgdorf, Germany) and 0.02 mg/kg midazolam (Midazolam Accord^®^, Accord Healthcare, Devon, UK) was used. After calming the animal and achieving intravenous access, induction for general anesthesia was performed using propofol (Propofol Lipuro^®^, B Braun, Melsungen AG, Melsungen, Germany) at a dose of 2 to 5 mg/kg, depending on the degree of sedation and the abolition of the larynx reflex. After induction and the application of additional local anesthesia to the larynx with lidocaine spray (Lidocain-Egis^®^, 10% solution, EGIS, Warsaw, Poland), the animal was intubated. Anesthesia was continued in the MRI room, using a special device for inhalation anesthesia that was allowed to work in a magnetic field (Philips, Dameca MRI 508, Eindhoven, The Netherlands). Inhalation anesthesia was carried out using isoflurane (flow 1.2–3.0 vol.%). Additional pain support was not necessary during imaging, following the procedures approved by the Local Ethics Committee in Wrocław. The constant monitoring of vital signs—heart rate (HR), saturation, capnography, inspiratory and expiratory carbon dioxide levels (PeCO_2_, EtCO_2_), and the number of breaths (RR—respiratory rate)—was conducted during the study using a cardiac monitor (Philips, Invivo Monitor, Expression MR400, Eindhoven, The Netherlands). After the examination, each animal was observed until it had completely awoken and stood up.

### 2.3. MR Imaging and DTI Protocol

The MR examinations were performed with a 1.5 Tesla magnetic resonance scanner (Philips, Ingenia) with 33 mT/m maximum gradient strength, using a sixteen-channel coil dedicated to head and spine imaging. The MR protocol consisted of sagittal T2-weighted images of the upper (TR/TE 3698/110 ms) and lower part of the spine (TR/TE 3698/120), followed by three axial DTI sequences.

DTI acquisition was based on single-shot spin-echo echo-planar imaging (SE/EPI) with the following parameters: cervical segment (TR/TE 5599/118 ms, 160 × 160 mm field of view, matrix 108 × 105 × 46 slices, voxel 1.5 × 1.5 mm with 3 mm thick axial slices); thoracolumbar segment (TR/TE 10150/118 ms, 180 × 180 mm field of view, matrix 120 × 118 × 80 slices, voxel 1.5 × 1.5 mm with 3 mm thick axial slices); and lumbar segment (TR/TE 6680/118 ms, 160 × 160 mm field of view, matrix 108 × 105 × 54 slices, voxel 1.5 × 1.5 mm with 3 mm thick axial slices). DTI was measured with an average directional resolution, i.e., in 15 diffusion directions. The SENSE factor was set between 1.5 and 3.0 [34]. The parameters of the DTI sequences were selected so that we could obtain reliable values for analysis without extending the time of anesthesia.

### 2.4. Image and DTI Analysis

The post-processing of the DTI data was performed using Philips DTI Fiber Track Software 2013 (Figure 2). The measurements were carried out in three sections: the cervical, thoracolumbar and lumbar segments of the spinal cord, exactly between the C4/C5, Th13/L1, and L4/L5 vertebrae. In each case, one segment was measured cranially and one segment caudally (Figure 3 and Figure 4). The ROIs were selected to represent the most vulnerable sites for spontaneous spinal cord injury.

The reference sagittal T2-weighted images of each evaluated level were used as an anatomical point.

The reconstruction of white matter tracts was performed by manually drawing the region of interest (in a linear shape) for three individual sections of the spinal cord. The ADC and FA metrics were measured on ADC and FA maps according to the manual placement of the nine abovementioned regions of interest (9 ROIs) in the center of the spinal cord in the midsagittal plane at the level of intervertebral disc spaces. ROIs were similar in size. Care was taken to position each ROI correctly, using structural images as an anatomical reference point to avoid the partial volume effect associated with the near presence of cerebrospinal fluid and bony structures (Figure 5). FA and ADC metrics were calculated and averaged over the selected voxels for each ROI [34].

### 2.5. Statistical Analysis

The collected values (FA and ADC) were subjected to statistical analysis. Testing for the normal distribution of the data obtained was performed with the Shapiro–Wilk normality test.

To assess the relationship between age, body weight, and the value of FA and ADC (for each of the 9 ROIs), a linear regression was used for the determination of the Pearson correlation coefficient, along with a test of statistical significance.

One-way analysis of variance (ANOVA) was used to compare the differences between the average FA and ADC values between nine ROIs for individual sections of the spinal cord. If the null hypothesis of equality of all means was rejected, we planned to use the post hoc cross-comparison test of Tukey’s HSD method.

The Statistica 13.3. data analysis software system (TIBCO Software Inc., Palo Alto, CA, USA, 2017) was used for statistical calculations, and *p* < 0.05 was considered statistically significant. We consulted with the university statistician regarding the selection of the statistical methodology.

## 3. Results

The means of the FA and ADC values for the three parts of the spinal cord and all regions of interest (ROIs) are shown in Table 1, Table 2 and Table 3 (DTI data from the Fiber Trak package).

A relationship was demonstrated between body weight, age, and FA and ADC values for the nine ROIs. It is noted that the FA-positive and ADC-negative values correlate with growth weight and age. One region showed no statistically significant correlation between body weight, age, and FA value (ROI4, thoracolumbar segment of the spinal cord). In the remaining cases, the existence of a parametric statistic was presented. The results of the conducted statistical analysis, including the value of the correlation coefficient and the significance level of the test, are shown in Figure 6, Figure 7, Figure 8, Figure 9, Figure 10 and Figure 11.

Comparing the differences between the mean FA and ADC values for each of the nine ROIs for the individual sections of the spinal cord (cervical, thoracolumbar and lumbar segments) showed no statistically significant differences (FA, *p* = 0.241, ADC, *p* = 0.462) (Figure 12 and Figure 13). Because the means did not differ significantly, we did not need to use the post hoc cross-comparison test of Tukey’s HSD method for further calculations.

## 4. Discussion

In the present study, we investigated whether pigs could be used as alternative model organisms for intact, growing spinal cords. We verified the fact proven by other researchers regarding the similarity in the neuroanatomical structure of the spinal cords of humans and pigs in comparison with other animal models such as monkeys, cats, and to a lesser degree, rats [35]. To our knowledge, this is the first study to report the usefulness of diffusion tensor parameters for assessing the microstructure of the healthy spinal cord in porcine.

Our research showed that there is a correlation between age, weight gain, and changes in FA and ADC parameters. The parameter features of FA and ADC at SCI sites have previously been identified: FA decreases and ADC increases [14,15,16,20]. We demonstrated that the opposite happens in the growth process: FA increases and ADC decreases. In the second part of the study, we established that the parameters of the diffusion tensor obtained in the porcine model were not equally significant in different parts (ROIs) of the individual sections of the spinal cord.

Other authors have described the changes in and relationships between DTI factors depending on age in healthy patient populations. The age groups of 20–77 years (*n* = 36) and 21–61 years (*n* = 65) show a significant dependence of FA on age, with the value decreasing across the whole cord. Moreover, it was shown that the obtained FA values decrease within grey matter compared to white matter (WM) in the aging process [27,31]. Another study on a healthy population aged 18–77 years (*n* = 36) confirmed that FA values decrease and ADC values increase with age [28]. This relationship is exactly the opposite of what we demonstrated in the spinal cord growth process. This shows that DTI is a useful method for spinal cord assessment, but the processes of maturation and aging need to be taken into account.

The duration of animals’ growth observation in our research was short (about 4 months, in one case 9.5 months). Pigs are fast-growing animals, which allowed us to obtain our results quickly. Due to the relatively short period of observation, the obtained data cannot be directly translated into the lifetime of companion animals or humans, but to the stage of development and maturation of the spinal cord, which in the cited human medicine literature, was observed for about twenty years. Therefore, we believe that in fast-growing animals, body weight is a better parameter to assess the growth process than age, but we included both the weight and age of the examined pigs in the analysis to better illustrate the rapid growth time of the selected animal model. In addition, the fact that in this study, we proved that changes in FA and ADC parameters with increasing body weight in pigs correspond to those obtained in humans during the growth and maturation process (depending on age), means, in our opinion, that body weight is a reliable parameter and reflects the growth process of fast-growing animals better than age. As described in the methodology section, the age of the animals was estimated and the weight was accurately measured. The analogical behavior of the FA and ADC parameters in both correlations shows that the results are comparable, but the dependence on the weight is more objective.

In the brain, the trajectory of DTI indices during maturation is well-described and characterized compared to the spinal cord, with increasing FA and other diffusivity factors—axial diffusivity (AD), radial diffusivity (RD), and mean diffusivity (MD)—decreasing. DTI has been shown to be a useful method for assessing myelination and microstructure changes in the brain during adolescence. It has been shown that DTI parameter charges can be used to evaluate which parts of the brain develop faster and which develop later. It has been proven that the GM/WM ratio in the brain decreases during the growth period. From birth to early childhood, the size of the brain increases rapidly, while at the later stages, development occurs by increasing the volume of WM and decreasing the volume of GM. This explains the behavior of DTI parameters in immaturity [36,37].

Changes in the values of the diffusion tensor in the growth process of the spinal cord in healthy pediatric patients have been described, but the concept is not as clear as in brain studies. Researchers also assessed the dependence on age in different groups: 6–16 years (*n* = 22), under 18 years of age (*n* = 41), 0.3–18 years (*n* = 121), and 6–16 years (*n* = 23) [30,34,38,39]. Most studies confirm the analogical dependence on age as the same as that found in this study in relation to age and body weight. FA values grow [30,34,38,39] and other parameters, such as ADC [34] or MD, become lower [30,39]. However, one study on a group of 121 pediatric patients did not confirm the previously described trends for MD and AD values in the maturing brain and spinal cord [38]. In most of the studies described, changes in the numerical quantity of the diffusion tensor during adolescence correlate with the ongoing myelination process and fiber packing, similar to that observed in the brain. As myelination progresses, the water content of the nervous tissue decreases, and the diameter of the nerve fiber thickens, which is also reflected in the DTI parameters [34]. It should be noticed that, unlike the brain, the spinal cord is myelinated in early infancy. The decrease in the GM/WM ratio in the growth process, also described in the brain, may not be the only phenomenon occurring in the spinal cord. A hypothesis has been proposed that with the growth of children, the proportion of larger axons increases, attempting to explain the differences in the obtained DTI values in the spinal cord growth process [38]. It is recognized that sequence parameters such as low SNR or the presence of cardiac and respiratory artifacts may affect the DTI parameters of the spinal cord and render the scope of changes unclear compared to the brain [10,29].

Our study proves that the parameters of the diffusion tensor obtained in the porcine model are not equally significant in different parts of the individual sections of the spinal cord. Nevertheless, looking at the obtained numerical values, a certain tendency can be noticed: FA decreases and ADC increases in the caudal direction of individual segments of the spinal cord (Figure 7 and Figure 8). The results of other researchers confirm the described trend, with FA values being particularly frequently used to evaluate differences in the parts of the cord, most often in the cervical spinal cord (CSC) [40,41,42,43,44,45,46,47,48]. Regional analysis showed that both the length of the spinal cord and the tract density grow in the maturation process. It has been noticed that the above-described phenomenon mostly affects the lower cervical and middle thoracic sections. It is estimated that, from the newborn stage to adulthood, the spinal cord grows approximately 2.7 times in length—the thoracic segment increases by 304%, and the cervical and lumbar by 238% [39]. The GM/WM ratio increases in the caudal core compared to the rostral level in adult and pediatric subjects. It should also be noted that the myelination process in the spinal cord is likely to be more mature and uniform in term newborns compared to the brain, which may explain the smaller regional difference in the obtained DTI values [38] The limitation of our analysis is the fact that it averages the FA and ADC values obtained from animals at different stages of growth, not taking into account the disproportion in growth in the length of the spine and spinal cord. The fact is known that the spinal column grows more rapidly in length compared to the spinal cord during growth and consequently, the spinal cord only extends to the lumbar spine in adult animals. This interesting aspect was also not included in other studies showing the relationship between MRI-DTI parameters on the course of the spinal cord or in its segments. The ambiguity of the results and the many variables affecting DTI values from the spinal cord imaging indicate the need to take into account the above-mentioned disproportion when planning future studies using this method to assess the microstructure of the spinal cord at various stages of its growth and maturation. However, it should be borne in mind that multivariate analysis may pose a significant interpretation challenge.

Pigs in the weight range of 24–120 kg participated in the examination. Polish White breed specimens reach a weight of 120 kg at the age of about 11 months. This is considered to be a late-maturing breed. Polish White pigs complete the growth process (mature) at the age of about 3 years, and the adult sow (female) reaches a body weight of 250–300 kg [49]. Relevant to the above-mentioned aspects, it should be noted that a 120 kg pig is still a growing animal, and the obtained results cannot be directly compared to those obtained from people of similar body weight. On the other hand, a pig this size is already fully developed and has reached reproductive maturity; therefore, it can be treated as a young adult. This age group can reflect young adults or animals with traumatic injuries to the spinal cord [2,14]. In the study, we showed the dynamics of a growing spinal cord. This analogy is confirmed by the results from humans during the growth period, depending on age [29,30,34,38,39,42].

It was possible to obtain DTI data from only one pig weighing 120 kg. This was due to the difficulty of conducting research procedures on an animal with such a large body weight. Veterinary service was complicated, and the number of anesthetic drugs used during the research process increased significantly. Based on the experience from the study, it was found that a weight of 60–65 kg would be the maximum, which is optimal for the assumed target age group. Even though the weight of 120 kg is a value that differs from the others, it was decided not to exclude it from the analysis because its removal did not change the statistical significance of the tests performed or the release of the analyzed parameters.

An interesting alternative to this study could be conducting an observation on minipigs (e.g., Göttingen) as an animal model. The Göttingen minipig is the smallest domestic pig in the world; as an adult, they weigh around 35 kg, which is much smaller than our target size group. The mentioned breed was specifically developed for biomedical research [50,51,52,53].

In our study, outcomes obtained from all swines were treated as separate results in the statistical analysis, although the animals participating in the examination had the same features—the same breed and breeding line, sex, as well as ways of keeping and feeding—and, therefore, had a similar daily weight gain. This is a guarantee of the obtained results’ comparability. Nevertheless, we believe that comparing more results from the same animal will be valuable for the complete imaging of the intact spinal cord model. We consider this fact as the aim of our research development.

To our knowledge, the results acquired on the animal model are more universal, homogeneous, and reproducible than those from single DTI study populations of people without spinal cord pathology in a specific age range. This is related to the great diversity of characteristics in the human community and shows yet another advantage resulting from research with the porcine model. It should also be mentioned that in the survey on the large breed (Polish White), we avoid the technical limitations related to the small size of the spinal cord, which is often described in studies on pediatric patients [29,30,34,38,39,42].

In our work, for the evaluation of the dynamics of changes in the spinal cord during the growth process, we used only two DTI parameters—FA and ADC—which could be considered a limitation. Meanwhile, other authors have shown the usefulness of assessing the microstructure of the spinal cord alongside the diffusion tensor and other parameters, for example, AD and MD, the second most useful parameters for evaluation in addition to FA and RD [27,31,38,39]. However, due to the wide range of applications and the described usefulness of FA [16,20,23] and ADC [6,19,23] in the assessment of spinal cord pathology, we decided to utilize them in this study.

Another limiting factor in our study was the fact that placement of the ROIs in the midsagittal plan does not allow for exact differentiation between GM and WM. Our methodology is derived from studies on pediatric patients [34] and was also reported in one study on a group of adults aged 18–77 years (*n* = 36) [28]. In neonates and very young children, the small size of the spinal cord did not allow the differentiation between GM and WM [34]. However, as previously explained in the pig model, we abolish the limitation related to the small size of the spinal cord, as we wanted to use a methodology suitable for use in humans during the period of growth. Nevertheless, we believe that there will be a chance for the development of this work and the selective measurement of ADC and FA values of grey or white matter within the spinal cord in the porcine model.

There are reports that the 3.0 Tesla field strength provides higher image quality in the DTI of the spinal cord compared to 1.5 T [54]. Carrying out the described procedures with the use of 3.0 T resonance and comparing the obtained outcomes would be an opportunity to improve our investigation.

## 5. Conclusions

Efforts to improve the quality of life of veterinary and human patients with SCIs require active research into new diagnostic methods. The outcomes of our large animal model on spinal cord advanced imaging fill the gap between experimental rodent studies and human studies. At the same time, we believe that these results are closer to being used in veterinary medicine than directly translating the relationships accompanying changes in FA and ADC parameters known from human medicine to companion animals. The presented large animal model obtains reference values that can be directly extrapolated to human medicine. Our study shows the particular usefulness of rapidly growing pigs as a model organism for the dynamic changes taking place in the spinal cord growth process for veterinary and human patients.

The described procedures and DTI planning protocol can be used directly in animals and humans in relation to their respective body weights and age. An assessment of the DTI structure with the use of FA and ADC parameters for traumatic spinal cord injuries should consider variations in spinal cord level in line with the organism’s growth time.

## Figures and Tables

**Figure 1 animals-13-00565-f001:**
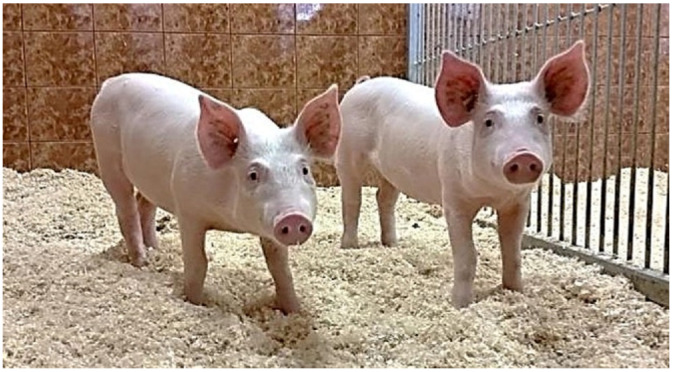
The picture shows the conditions the animals were kept in.

**Figure 2 animals-13-00565-f002:**
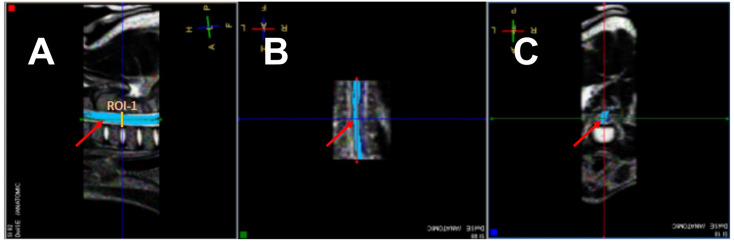
Post-processing of the DTI data (Philips DTI Fiber Track Software 2013). Location of the first region of interest (ROI1) in the cervical segment of the spinal cord. (**A**)–sagittal plane, (B)–dorsal plane, (**C**)–transverse plane. The red arrow marks the reconstruction of the white matter tract (tractography) in individual planes A–C.

**Figure 3 animals-13-00565-f003:**
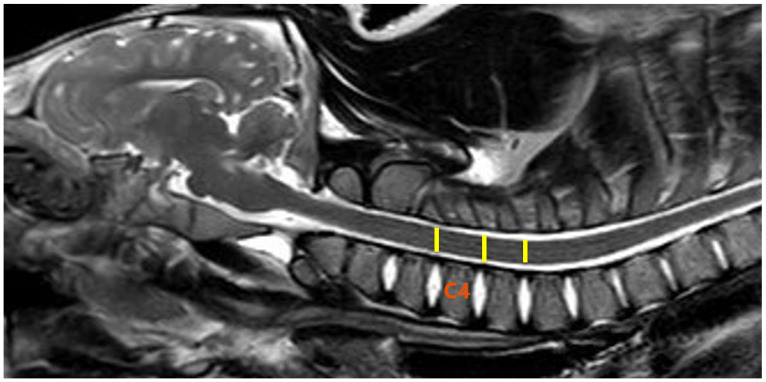
Location of regions of interest (ROIs) in the cervical segment of the spinal cord (yellow lines). T2-weighted sagittal image of the pig from the research group. Fourth cervical vertebra as a reference point (C4).

**Figure 4 animals-13-00565-f004:**
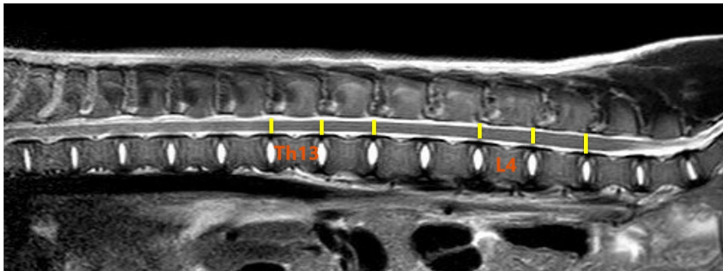
Location of ROIs in the thoracolumbar segment of the spinal cord (yellow lines). T2-weighted sagittal image of the pig from the research group. Thirteenth thoracic vertebrae (Th13) and fourth lumbar vertebrae (L4) as reference points.

**Figure 5 animals-13-00565-f005:**
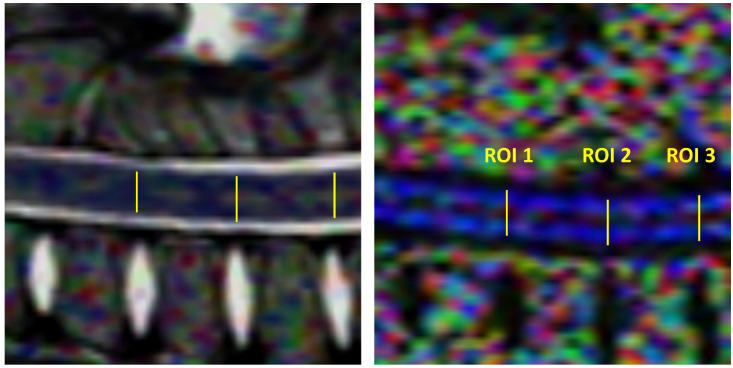
Sagittal T2-weighted MR image and FA map of the cervical spinal cord showing ROIs placement. In the other segments, via analogy.

**Figure 6 animals-13-00565-f006:**
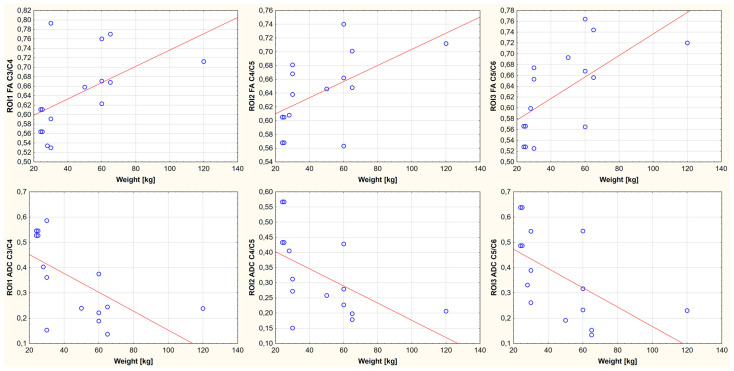
Linear fit plots show the relationship between FA (the top row) and ADC (the bottom row) values and weight for the cervical segment of the spinal cord. Top row from the left: ROI1 (*p*-value: 0.040, r value: 0.535), ROI2 (*p*-value: 0.028, r value: 0.565), ROI3 (*p*-value: 0.015, r value: 0.644). Bottom row from the left: ROI1 (*p*-value: 0.034, r value: −0.548) ROI2 (*p*-value: 0.034, r value: −0.548), and ROI3 (*p*-value: 0.023, r value: −0.579).

**Figure 7 animals-13-00565-f007:**
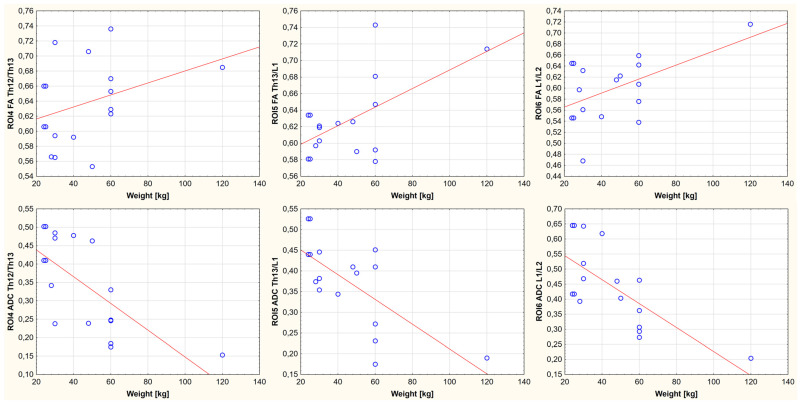
Linear fit plots show the relationship between FA (the top row) and ADC (the bottom row) values and weight for the thoracolumbar segment of the spinal cord. Top row from the left: ROI4 (*p*-value: 0.168, r value: 0.350), ROI5 (*p*-value: 0.014, r value: 0.582), ROI6 (*p*-value: 0.033, r value: 0.519). Bottom row from the left: ROI4 (*p*-value: 0.002, r value: −0.695) RO5 (*p*-value: 0.002, r value: −0.695), and ROI6 (*p*-value: 0.002, r value: −0.708).

**Figure 8 animals-13-00565-f008:**
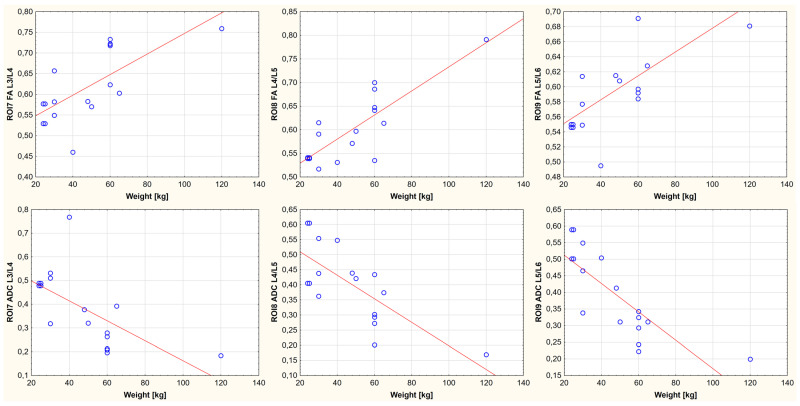
Linear fit plots show the relationship between FA (the top row) and ADC (the bottom row) values and weight for the lumbar segment of the spinal cord. Top row from the left: ROI7 (*p*-value: 0.002, r value: 0.698), ROI8 (*p*-value < 0.001, r value: 0.829), ROI9 (*p*-value: 0.002, r value: 0.694). Bottom row from the left: ROI7 (*p*-value: 0.005, r value: −0.650) ROI8 (*p*-value: 0.001, r value: −0.734), and ROI9 (*p*-value < 0.001, r value: −0.805).

**Figure 9 animals-13-00565-f009:**
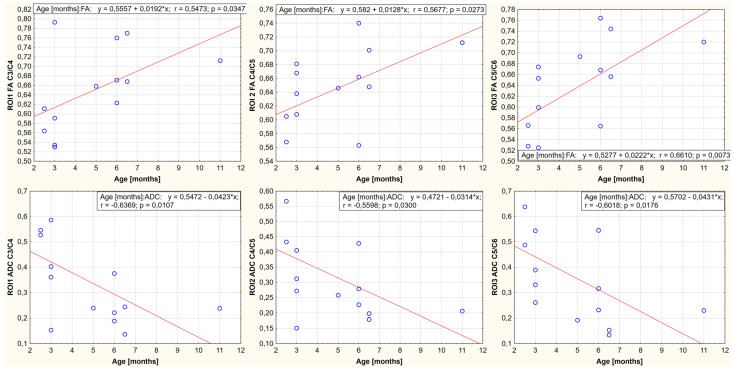
Linear fit plots show the relationship between FA (the top row) and ADC (the bottom row) values and age for the cervical segment of the spinal cord. Top row from the left: ROI1 (*p*-value: 0.035, r value: 0.547), ROI2 (*p*-value: 0.027, r value: 0.568), ROI3 (*p*-value: 0.007, r value: 0.661). Bottom row from the left: ROI1 (*p*-value: 0.010, r value: −0,637), ROI2 (*p*-value: 0.030, r value: −0.560), and ROI3 (*p*-value: 0.018, r value: −0.602).

**Figure 10 animals-13-00565-f010:**
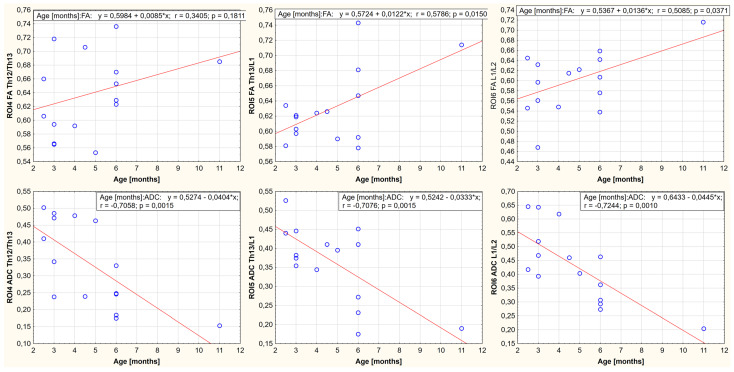
Linear fit plots show the relationship between FA (the top row) and ADC (the bottom row) values and age for the thoracolumbar segment of the spinal cord. Top row from the left: ROI4 (*p*-value: 0.181, r value: 0.340), ROI5 (*p*-value: 0.015, r value: 0.579), ROI6 (*p*-value: 0.037, r value: 0.509). Bottom row from the left: ROI4 (*p*-value: 0.0015, r value: −0.706), RO5 (*p*-value: 0.0015, r value: −0.708), and ROI6 (*p*-value: 0.001, r value: −0.724).

**Figure 11 animals-13-00565-f011:**
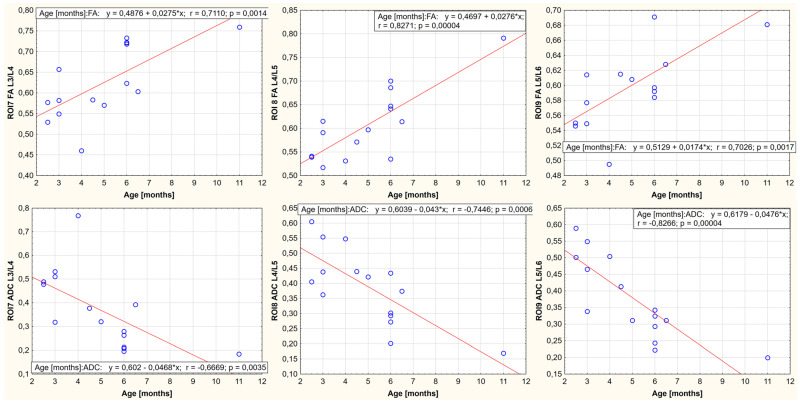
Linear fit plots show the relationship between FA (the top row) and ADC (the bottom row) values and age for the lumbar segment of the spinal cord. Top row from the left: ROI7 (*p*-value: 0.001, r value: 0.711), ROI8 (*p*-value < 0.001, r value: 0.827), ROI9 (*p*-value: 0.001, r value: 0.703). Bottom row from the left: ROI7 (*p*-value: 0.0035, r value: −0.667), ROI8 (*p*-value: < 0.001, r value: −0.745), and ROI9 (*p*-value < 0.001, r value: −0.827).

**Figure 12 animals-13-00565-f012:**
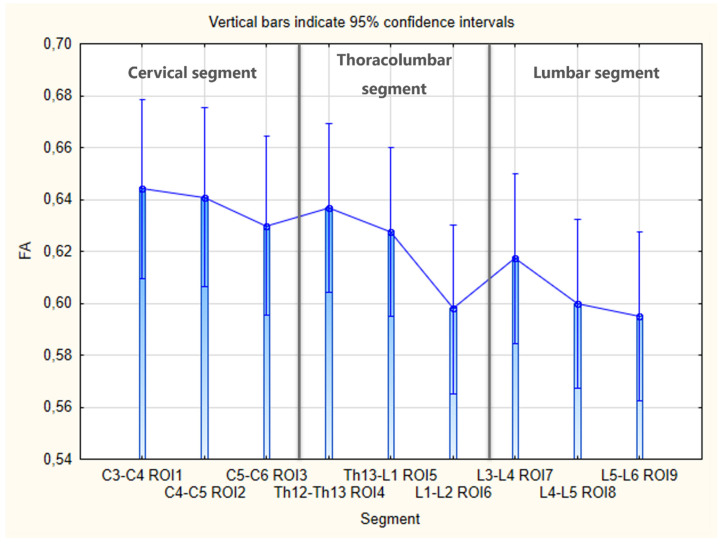
Bar graph with boxplots comparing the differences in the mean FA values between each of the nine ROIs for individual sections of the spinal cord; One-way analysis of variance test (ANOVA test), *p* = 0.241.

**Figure 13 animals-13-00565-f013:**
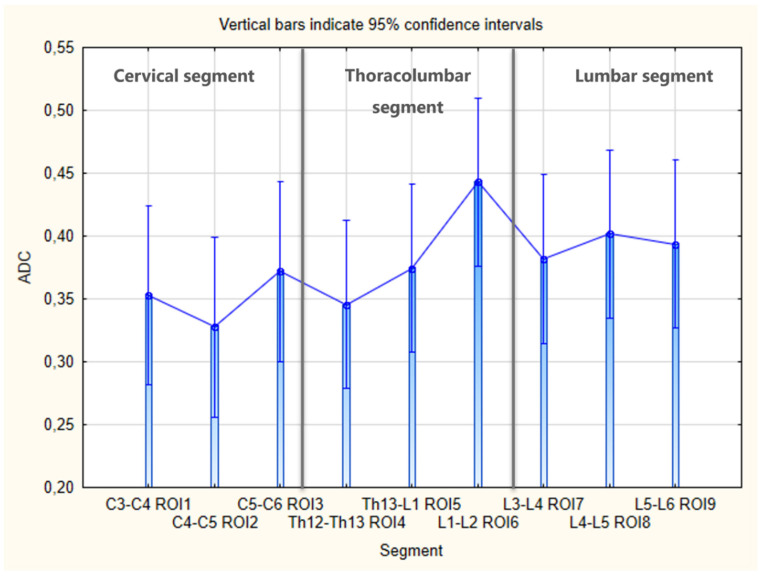
Bar graph with boxplots comparing the differences in the mean ADC values between each of the nine ROIs for individual sections of the spinal cord; ANOVA test, *p* = 0.462.

**Table 1 animals-13-00565-t001:** Mean fractional anisotropy (FA) and apparent diffusion coefficient (ADC) values of ROIs in the cervical segment of the spinal cord.

Porcine	Age (Months)	Weight (kg)	Cervical Segment					
			ROI1		ROI2		ROI3	
			FA C3/C4	ADC C3/C4	FA C4/C5	ADC C4/C5	FA C5/C6	ADC C5/C6
1	2.5	24	0.611	0.546	0.605	0.433	0.528	0.638
2	2.5	24	0.564	0.527	0.568	0.567	0.566	0.487
3	2.5	25	0.564	0.527	0.568	0.567	0.566	0.487
4	2.5	25	0.611	0.546	0.605	0.433	0.528	0.638
5	3	28	0.534	0.403	0.608	0.405	0.599	0.331
6	3	30	0.591	0.361	0.681	0.151	0.653	0.389
7	3	30	0.53	0.586	0.668	0.272	0.525	0.544
8	3	30	0.793	0.153	0.638	0.312	0.674	0.261
9	4	40						
10	4.5	48						
11	5	50	0.658	0.239	0.646	0.258	0.693	0.192
12	6	60						
13	6	60	0.623	0.375	0.563	0.428	0.565	0.545
14	6	60						
15	6	60	0.671	0.221	0.662	0.279	0.668	0.316
16	6	60	0.76	0.189	0.74	0.227	0.764	0.232
17	6.5	65	0.668	0.244	0.701	0.198	0.744	0.153
18	6.5	65	0.77	0.137	0.648	0.179	0.656	0.134
19	11	120	0.712	0.238	0.712	0.206	0.72	0.23

**Table 2 animals-13-00565-t002:** Mean FA and ADC values of ROIs in the thoracolumbar segment of the spinal cord.

Porcine	Age (Months)	Weight (kg)	Thoracolumbar Segment					
			ROI4		ROI5		ROI6	
			FA Th12/Th13	ADC Th12/Th13	FA Th13/L1	ADC Th13/L1	FA L1/L2	ADC L1/L2
1	2.5	24	0.606	0.502	0.581	0.526	0.645	0.417
2	2.5	24	0.66	0.41	0.634	0.44	0.546	0.645
3	2.5	25	0.66	0.41	0.634	0.44	0.546	0.645
4	2.5	25	0.606	0.502	0.581	0.526	0.645	0.417
5	3	28	0.566	0.342	0.597	0.374	0.597	0.393
6	3	30	0.565	0.485	0.621	0.446	0.561	0.468
7	3	30	0.718	0.238	0.603	0.382	0.468	0.643
8	3	30	0.594	0.471	0.619	0.354	0.632	0.519
9	4	40	0.592	0.478	0.624	0.344	0.548	0.618
10	4.5	48	0.706	0.239	0.626	0.41	0.615	0.46
11	5	50	0.553	0.463	0.59	0.395	0.622	0.403
12	6	60	0.736	0.175	0.743	0.175	0.659	0.273
13	6	60	0.629	0.248	0.592	0.41	0.642	0.307
14	6	60	0.623	0.33	0.647	0.272	0.607	0.293
15	6	60	0.653	0.246	0.681	0.231	0.576	0.463
16	6	60	0.67	0.184	0.578	0.451	0.538	0.362
17	6.5	65						
18	6.5	65						
19	11	120	0.685	0.153	0.714	0.19	0.716	0.204

**Table 3 animals-13-00565-t003:** Mean FA and ADC values of ROIs in the lumbar segment of the spinal cord.

Porcine	Age (Months)	Weight (kg)	Lumbar Segment					
			ROI7		ROI8		ROI9	
			FA L3/L4	ADC L3/L4	FA L4/L5	ADC L4/L5	FA L5/L6	ADC L5/L6
1	2.5	24	0.577	0.478	0.539	0.405	0.546	0.589
2	2.5	24	0.529	0.488	0.541	0.605	0.55	0.501
3	2.5	25	0.529	0.488	0.541	0.605	0.55	0.501
4	2.5	25	0.577	0.478	0.539	0.405	0.546	0.589
5	3	28						
6	3	30	0.582	0.531	0.591	0.438	0.577	0.465
7	3	30	0.549	0.51	0.517	0.554	0.549	0.549
8	3	30	0.657	0.318	0.615	0.362	0.614	0.338
9	4	40	0.46	0.768	0.531	0.548	0.495	0.504
10	4.5	48	0.583	0.377	0.571	0.439	0.615	0.413
11	5	50	0.57	0.32	0.597	0.421	0.608	0.311
12	6	60	0.721	0.196	0.686	0.293	0.691	0.243
13	6	60	0.722	0.213	0.7	0.201	0.691	0.222
14	6	60	0.623	0.279	0.647	0.272	0.584	0.293
15	6	60	0.718	0.208	0.641	0.302	0.592	0.342
16	6	60	0.733	0.263	0.535	0.434	0.597	0.324
17	6.5	65						
18	6.5	65	0.603	0.392	0.614	0.374	0.628	0.311
19	11	120	0.759	0.184	0.791	0.169	0.681	0.199

## Data Availability

The data presented in this study are available on request from the corresponding author.

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
