# Peer review of "Porcine Model of the Growing Spinal Cord—Changes in Diffusion Tensor Imaging Parameters"

_animals, 2023, doi:10.3390/ani13040565_

Round 1

Reviewer 1 Report

The manuscript shows detailed informations about MRI parameters of the growing spinal cord in pigs using DTI.

Overall, the authors provide interesting data for the field of CNS development and MRI.

Some aspects of the manuscript have to be reconsidered:

- the relevance of the measured MRI values in uninjured animals for SCI is not clear from reading the manuscript

- It is necessary to emphasize the novelty of the data compared to the data already published and cited

- Figures 7 and 8 should be displayed differently, e. g. as a bar graph

Author Response

Response to Reviewer 1 Comments
Point 1: The relevance of the measured MRI values in uninjured animals for SCI is not clear from reading the manuscript
Response 1:
The aim of our study was to assess the DTI values in the process of growth and depending on the individual segments of the spinal cord. As shown in the study assessing spinal cord in the young organism is different from older ones.
The pig is a well-known experimental animal model used in translational medicine, also as a model of damaged spinal cord (ref.: A) Mercedes Zurita, Concepción Aguayo, Celia Bonilla, Laura Otero, Miguel Rico, Alicia Rodríguez, Jesús Vaquero, The pig model of chronic paraplegia: A challenge for experimental studies in spinal cord injury, Progress in Neurobiology, Volume 97, Issue 3, 2012, Pages 288-303, https://doi.org/10.1016/j.pneurobio.2012.04.005.; B) Weber-Levine, C., Hersh, A. M., Jiang, K., Routkevitch, D., Tsehay, Y., Perdomo-Pantoja, A., Judy, B. F., Kerensky, M., Liu, A., Adams, M., Izzi, J., Doloff, J. C., Manbachi, A., & Theodore, N. (2022). Porcine Model of Spinal Cord Injury: A Systematic Review. Neurotrauma reports, 3(1), 352–368. https://doi.org/10.1089/neur.2022.0038.; C) Rakib Uddin 
Ahmed, Chase A. Knibbe, Felicia Wilkins, Leslie C. Sherwood, Dena R. Howland, Maxwell Boakye, Porcine spinal cord injury model for translational research across multiple functional systems, Experimental Neurology, Volume 359, 2023, https://doi.org/10.1016/j.expneurol.2022.114267.).

In our study, we wanted to obtain the reference DTI values from healthy spinal cord, so that in subsequent works, we would refer directly to them obtained in the porcine SCI experimental animal model. We also wanted to take advantage of the fact that DTI, unlike standard structural MRI sequences, allows for a more 
objective assessment of the degree of spinal cord damage or its regeneration after treatment. For these promising assumptions, we wanted to test the usefulness of the DTI method in this animal model.
In our opinion, DTI is a very promising method, however it is quite a new technique for clinical use. Obtaining objective reference values is necessary for further clinical trials, to have a reference point on the values obtained from clinical patients with SCI - we added this sentence to the manuscript.
The primary goal of our work was to better understand the DTI technique and the variables that may influence ‘the behavior’ of the FA and ADC parameters in the uninjured spinal cord. In our opinion, without reference values and without understanding the influence of factors, like the growing process - expressed in our research by body weight, it is impossible to objectively assess the injured 
spinal cord by this advanced MRI technique.

The relevance of our research for future studies and for animals and humans with injured spinal cord is specified in Manuscript:
• The purpose of this study was to emphasize the benefits of creating and optimizing animal translational models. The results are thought to contribute to the determination of DTI reference values for pigs with intact spinal cords and could be used for comparison with the ones obtained from porcine models of SCI in further studies. The study verifies the pig as a suitable model for translational medicine in this field. (Manuscript 90-94)

• DTI is a very promising method, however it is quite a new technique for clinical use. Obtaining objective reference values is necessary for further clinical trials, to have a reference point on the values obtained from clinical patients with SCI. (Manuscript 94-97)
Efforts to improve the quality of life of veterinary and human patients with SCIs require active research into new diagnostic methods. The outcomes of our large animal model on spinal cord advanced imaging fill the gap between experimental rodent studies and human studies. At the same time, we believe that these results are closer to being used in veterinary medicine than directly translating the relationships accompanying changes in FA and ADC parameters known from human medicine, to companion animals. The presented large 
animal model obtains reference values that can be directly extrapolated to human medicine. Our study shows the particular usefulness of rapidly growing pigs as a model organism for the dynamic changes taking place in the spinal cord growth process for veterinary and human patients. (Manuscript 400-409)
Point 2: It is necessary to emphasize the novelty of the data compared to the data already published and cited.
Response 2:
In our research pig was used as an alternative animal model for veterinary and human medicine. Most often a rat model of an intact or damaged spinal cord is reported. According to other research and our experiences, the pig animal model is more similar to human beings than the experimental rodent model. Research in laboratory animal models has its limitations. Results obtained in studies 
on rats may not fully reflect the health problems which affected the companion animals and humans - that is the first novelty of our study on pigs. Second, we have conducted the study about assessing the maturation of a spinal cord and differences between the individual segments of the spinal cord using the DTI technique - from our knowledge, it is the first study conducted in this way in veterinary medicine on this animal model.
Moreover, the size of this pig animal model allows the use of the same imaging tools, scanners, software, and interpretation used in everyday clinical and advanced imaging sets. Despite these advantages, there are no reports of the use of these animals as a model for undamaged, growing spinal cords, which proves the novelty of our research. In our opinion, the above-mentioned aspects 
show that the pig is an universal model for veterinary and human medicine, and our results fill the gap between experimental rodent studies and human studies. At the same time, we believe that these results are closer to being used in veterinary medicine than directly translating the relationships accompanying changes in FA and ADC parameters known from human medicine, to companion 
animals - we added this sentence to the manuscript.

We emphasize novelty of our study in the Manuscript:
Moreover, the size of this animal model allows the use of the same imaging tools, scanners, software, and interpretation used in everyday clinical and advanced imaging sets. Despite these advantages, there are no reports of the use of these animals as a model for undamaged, growing spinal cords, which proves the novelty of our research. (Manuscript 78-82)
In the present study, we investigated whether pigs could be used as alternative model organisms for intact, growing spinal cords. We verified the fact proven by other researchers regarding the similarity in the neuroanatomical structure of the spinal cords of humans and pigs, in comparison with other animal models such as monkeys, cats, and to a lesser degree, rats. To our knowledge, this is the first study examining the usefulness of diffusion tensor parameters for assessing the microstructure of the healthy spinal cord in porcine.(Manuscript 253-259)
The outcomes of our large animal model on spinal cord advanced imaging fill the gap between experimental rodent studies and human studies. At the same time, we believe that these results are closer to being used in veterinary medicine than directly translating the relationships accompanying changes in FA and ADC parameters known from human medicine, to companion animals. (Manusript 401-405)

Point 3: Figures 7 and 8 should be displayed differently, e. g. as a bar graph
Response 3:
Thank You for Your suggestion, we have changed it in the Manuscript. We have changed the numbering of figures (Figures 9 and 10)

Reviewer 2 Report

The work presented here by Owsińska-Schmidt and colleagues is based on an interesting idea to establish the reference values from commonly used DTI indices such as FA and ADC in an intact porcine spinal cord. Overall, the work presented here is well described, and clearly communicated. However, there are a few issues that need to be addressed.

The title mentions “growing” spinal cord, however, the authors have only considered weight as an indicator of growth. While the correlation between growth and weight gain is well established in porcine and several other animal models, there are also known caveats to this. I strongly suggest that the age factor needs to be addressed here. Additionally, it is my opinion that the total length of spinal column would be a better indicator of growth than using just the body weight. Especially to increase the relevance of the discussion points raised in lines 305-307 regarding the increase in spinal cord length during paediatric growth. Thus, the grouping of animals should also be done by the age and length of spinal column. It would also be useful if the authors can then draw parallels or explain equivalence between the ages of the included pigs with human growth periods in the discussion section.

Another critical aspect that must be addressed is the length differential between the spinal cord and spinal column that changes with growth. Adding the age groups to the analysis criteria will also help with this. Since the ROIs were determined based on vertebral landmark, it must be noted that they will not have coincided with the same spinal cord segments across different age groups, and the cervical and lumbosacral enlargements in the cord would not have been considered during analysis. At birth, the spinal cord length is close to spinal column length – meaning that the vertebral levels coincide with the spinal cord segments quite closely. However, the spinal column grows faster and more in length compared to the spinal cord during post-natal growth and therefore, the spinal cord only extends till the lumbar spine in adult animals.  This must be accounted for in the methodology, analysis and discussion aspects.

As a result, Figures 4, 5 and 6 can be modified to display the data against the age and spinal column length on the x-axis. Similarly, the data in figures 7 and 8 should be split based on the groupings to depict if there were any statistically significant differences observed against the animal’s growth.

For results, if the authors could also present the reconstructed MR images from all 9 ROIs from the representative porcine models, showing the cross sections at the different ROI levels, that can provide better context to the numerical values provided here. This will also help address the cervical and lumbosacral enlargements in the spinal cords which alter the grey matter/white matter proportions that would in turn likely affect the FA and ADC.

Thus, the manuscript overall represents good data collection, but the analysis and interpretation of the same overlooks some critical points, decreasing the scientific rigor of the manuscript as a whole. These points must be considered and the data must be re-interpreted accordingly.

Author Response

Response to Reviewer 2 Comments
Point 1: The title mentions “growing” spinal cord, however, the authors have only considered weight as an indicator of growth. While the correlation between growth and weight gain is well established in porcine and several other animal models, there are also known caveats to this. I strongly suggest that the age factor needs to be addressed here.
It would also be useful if the authors can then draw parallels or explain equivalence between the ages of the included pigs with human growth periods in the discussion section.
Response 1:
Thank you very much for the interesting and developing review.
The duration of animals growth observation in our research was short (about 4 months) - this fact is emphasized in the Manuscript. We used the advantage of the fact that pigs are fast-growing animals and it allowed us to get results quickly. Due to the relatively short period of observation, the obtained 
data cannot be directly translated into the lifetime of companion animals or humans, but to the stage of development and maturation of the spinal cord, which in the cited human medicine literature was observed for about twenty years. Therefore, we believe that in fast-growing animals, body weight is 
a better parameter to assess the growth process than age. In addition, the fact that in the study we proved that changes in FA and ADC parameters with increasing body weight in pigs correspond to those obtained in humans during the growth and maturation process (depending on age) means in our opinion that body weight is a reliable parameter and better reflects growth process of fastgrowing animals than age. - added an explanatory paragraph to the discussion section.

The 4-monthly observations of the pig model growing spinal cord process resemble roughly twenty years of human growth. - we removed the above sentence from the introduction section as it could lead to a misleading interpretation.
We explain aspects mentioned by the reviewer in the Manuscript:
• We decided to use pigs without SCI as a model organism due to their rapid weight gain and anatomical and physiological similarity to humans. These features allow quick and reliable results with different sizes during the growth of the animal’s spinal cord, forming an excellent model for translational SCI research in the fields of veterinary and human medicine. (Manuscript 72-76)
• Piglets of the same breed (Polish White) with similar body weight, i.e., approx. 15–20 kg, were used in the study. The animals were observed for about 4 months. (Manuscript 100-101)
The duration of animals growth observation in our research was short (about 4 months). We used the advantage of the fact that pigs are fast-growing animals and it allowed us to get results quickly. Due to the relatively short period of observation, the obtained data cannot be directly translated into the lifetime of companion animals or humans, but to the stage of development and maturation of the spinal cord, which in the cited human medicine literature was observed for about twenty years. Therefore, we believe that in fast-growing animals, body weight is a better parameter to assess the growth process than age. In addition, the fact that in the study we proved that changes in FA and ADC parameters with increasing body weight in pigs correspond to those obtained in humans during the growth and maturation process (depending on age) means in our opinion that body weight is a reliable parameter and better reflects growth process of fast-growing animals than age. (Manuscript 277-288)
Pigs in the weight range of 24–120 kg participated in the examination. Polish White breed specimens reach a weight of 120 kg at the age of about 6 months. This is considered to be a late-maturing breed. Polish White pigs complete the growth process (mature) at the age of about 3 years, and the adult sow (female) reaches a body weight of 250–300 kg. Relevant to the above-mentioned aspects, it should be noted that a 120 kg pig is still a growing animal, and the obtained results cannot be directly compared to those obtained from people of similar body weight. On the other hand, this size pig is already fully developed and has 
reached reproductive maturity; therefore, it can be treated as a young adult. This age group can reflect young adults or animals with traumatic injuries to the spinal cord. In the study, we showed the dynamics of a growing spinal cord. This analogy is confirmed with the results from humans during the growth period, depending on age. (Manuscript 336-347)
Point 2: Additionally, it is my opinion that the total length of spinal column would be a better indicator of growth than using just the body weight. Especially to increase the relevance of the discussion points raised in lines 305-307 regarding the increase in spinal cord length during paediatric growth. Thus, the grouping of animals should also be done by the age and length of spinal column.
Another critical aspect that must be addressed is the length differential between the spinal cord and spinal column that changes with growth. Adding the age groups to the analysis criteria will also help with this. Since the ROIs were determined based on vertebral landmark, it must be noted that they will not have coincided with the same spinal cord segments across different age groups, and the cervical and lumbosacral enlargements in the cord would not have been considered during analysis. 
At birth, the spinal cord length is close to spinal column length – meaning that the vertebral levels coincide with the spinal cord segments quite closely. However, the spinal column grows faster and more in length compared to the spinal cord during post-natal growth and therefore, the spinal cord only extends till the lumbar spine in adult animals. This must be accounted for in the methodology, analysis and discussion aspects.
Response 2:
Thank you very much for Your interesting opinion.
Due to the duration of the project and local ethical committee approval, we were not able to perform an examination along the entire length of the spine in all 19 examination pigs. The data comes from control studies of animals involved in the experimental spinal cord injury project. The sequences were selected to optimize the animals' anesthesia time. In addition, the data of most of them were analyzed retrospectively, so we cannot reconstruct the whole spinal cord, and spinal column length from all of our research material, to ensure the appropriate statistical power of the analysis of this aspect. However, this is a very interesting point of view. We believe that this is a good direction for the development of our research in the future. In this work, however, we would like to focus on the 
discussed aspects of the dependence of diffusion tensor parameters on body weight as a determinant of animal growth, and the relationship of FA and ADC parameters in individual sections of the spinal cord.

The ROIs were selected to represent the most vulnerable sites for spontaneous spinal cord injury. At the same time, a compromise was required between the duration of the MRI examination and the optimization of the quality of the obtained data, and the time of general anesthesia in the tested animals.

The reviewer makes a very interesting consideration about the differences in growth in the length of the spine and the spinal cord. We believe that this aspect should be taken into account in future studies using the diffusion tensor technique to assess the spinal cord, especially in growing patients.
However, it should be borne in mind that despite these differences in growth time, the intervertebral spaces are the only reliable reference point for determining neurolocalization both in classical radiology and with the use of advanced MRI techniques.

Point 3: As a result, Figures 4, 5 and 6 can be modified to display the data against the age and spinal column length on the x-axis. Similarly, the data in figures 7 and 8 should be split based on the groupings to depict if there were any statistically significant differences observed against the animal’s growth.
Response 3:
No changes have been made at the moment.

Point 4: For results, if the authors could also present the reconstructed MR images from all 9 ROIs from the representative porcine models, showing the cross sections at the different ROI levels, that can provide better context to the numerical values provided here. This will also help address the cervical and lumbosacral enlargements in the spinal cords which alter the grey matter/white matter proportions that would in turn likely affect the FA and ADC.
Response 4:
We added new pictures in a Manuscript to better illustrate our methodology (Figure 2 and Figure 5).
In our work, we decided to use a less frequently common, but equally well-described method of obtaining diffusion tensor parameter values by overlapping regions of interest in the sagittal plane of the spinal cord.
Although this methodology does not allow for a precise differentiation of the obtained DTI parameters between white and gray matter, it seems to be particularly useful for the assessment of the spinal cord of pediatric patients and small animals (up to 20 kg). In this group of patients, due to the small size of the spinal cord, the differentiation between gray and white matter using the 1.5 T 
resonance imaging scanner has a significant limitation.
We have explained in more detail the reasons for using the above-mentioned method and its limitations in the Manuscript:
Another limiting factor in our study was the fact that placement of the ROIs in the midsagittal plan does not allow for exact differentiation between GM and WM. Our methodology is derived from studies on pediatric patients and was also reported in one study on a group of adults aged 18–77 years (n = 36). In neonates and very young children, the small size of the spinal cord did not allow the differentiation between GM and WM. Although, as previously explained in the pig model, we abolish the limitation related to the small size of the spinal 
cord, as we wanted to use a methodology suitable for use in humans during the period of growth. Nevertheless, we believe that there will be a chance for the development of this work and the selective measurement of ADC and FA values of grey or white matter within the spinal cord in the porcine model. (Manuscript 385-394)
• There are reports that the 3.0 Tesla field strength provides higher image quality in the DTI of the spinal cord compared to 1.5 T. Carrying out the described procedures with the use of 3.0 T resonance and comparing the obtained outcomes would be an opportunity to improve our investigation. (Manuscript 395-398)

Round 2

Reviewer 1 Report

The points have been addressed accordingly.

Author Response

Thanks to the Reviewer for the response and previous comments.

Reviewer 2 Report

The authors Owsińska-Schmidt and colleagues have addressed several of the comments very well. I am very happy with the addition/editing of the figures with expressed clarifications in the methodology as well as discussion regarding points 1 and 4 raised in my previous comments.

However, the points 2 and 3 from earlier, yet remain to be satisfactorily addressed. I can appreciate the time limitations and ethics considerations, and also accept that certain aspects may not be feasible to be actioned upon at this time.

However, the rationale of this study is to generate reference values for any and all future animal or clinical trials to enable the investigators to better utilise the DTI techniques. This places a significant burden on the authors and this crucial work to have better, more robust presentation of the gathered data. There are aspects that the future trials can address and improve upon, as it is well indicated by the authors. Nevertheless, I draw the authors' attention to this particular comment from point 2:

"Another critical aspect that must be addressed is the length differential between the spinal cord and spinal column that changes with growth. Adding the age groups to the analysis criteria will also help with this. Since the ROIs were determined based on vertebral landmark, it must be noted that they will not have coincided with the same spinal cord segments across different age groups, and the cervical and lumbosacral enlargements in the cord would not have been considered during analysis. At birth, the spinal cord length is close to spinal column length – meaning that the vertebral levels coincide with the spinal cord segments quite closely. However, the spinal column grows faster and more in length compared to the spinal cord during post-natal growth and therefore, the spinal cord only extends till the lumbar spine in adult animals. This must be accounted for in the methodology, analysis and discussion aspects."

While I completely agree with the authors that the vertebral landmarks represent the most reliable and reproducible way to generate and define the ROIs, the trade-off of the same as pointed out in the comment must still be addressed. It is not clear why the same MR images cannot be used to calculate the length of the spine and the spinal cord in the mid sagittal plane. Even if it is not possible to do so, the age-wise correlation of the spinal segments with the vertebral levels can still be acquired from historical works and the secondary analysis can be included in this manuscript. This will not require access to the original animals, only the imaging data which is already collected. 

Finally, I would like to congratulate the authors on this fundamental work and the tremendous efforts poured into it. Addressing these comments will raise the quality and impact of this work further to meet its intended rationale. 

Author Response

Point 4 and 5, and also 2 and 3 from the earlier review:

Authors Response:

Thanks again to the Reviewer for a thoughtful and inspiring review.

Due to the Reviewer's suggestion to include the age categories of animals in our research, we hope to better show the dynamics of changes in DTI parameters in the growth process and emphasize the advantages of using a fast-growing animal model in our study.

However, the suggested changes required us to plan and conduct new statistical analyzes of the obtained results. We would like to ask the Publishing Company for the possibility of extending the time to make the necessary changes to the manuscript, in relation to the Reviewer's suggestions.

We would like our results to be presented in a consistent manner, so we need more time to make the necessary changes to the manuscript, summarize the newly performed analyzes and modify the discussions section.

Round 3

Reviewer 2 Report

If the authors are allowed more time to make the changes, I would be happy to review the changes. 

Author Response

Thank You for all Your valuable comments that allowed us to look at our results in a completely different way.

In order to better illustrate the growth process of the spinal cord on the porcine model, we extended the analysis of changes in the MRI-DTI values depending on the age of the tested animals. We obtained an analogous relationship of parameters as in the case of dependence on body weight growth.

In the analysis of differences between DTI values in the individual ROIs along the course of the spinal cord, we emphasized the limitations from disregard resulting in the disproportion between the growth of the spine and the spinal cord in the growth process. This interesting aspect, emphasized by You, was also not included in other studies showing the relationship between MRI-DTI parameters on the course of the spinal cord or in its individual segments. The ambiguity of the results and the many variables affecting DTI values from the spinal cord imaging, indicate the need to take into account the above-mentioned disproportion when planning future studies using this method to assess the microstructure of the spinal cord at various stages of its growth and maturation. However, it should be borne in mind that multivariate analysis may pose a significant interpretation challenge - we added this paragraph to the manuscript.

Ultimately, we decided not to analysis the differences between individual segments, taking into account the divisions between intra classes determined by age/weight groups. This very interesting comparison made it possible to take into account indirectly the disproportions mentioned above. However, the division into smaller groups (depending on age/weight) in our study on 19 pigs meant that they were small in number and the statistical tests used were characterized by lower power, which introduced new limitations on interpretation of the obtained data. Such comparisons will be possible in the future when a larger research group will be planned.

Due to the retrospective nature of our work, we were unable to obtain the values of length of the spinal cord and spine from animals at individual growth stages, because our imaging was limited to areas covering the ROIs, due to the need to shorten the time of the examination under general anesthesia as much as possible. However, we would like to develop our future research for a better understanding of the relationship of DTI parameters in spinal cord assessment.